# Involvement of Akt/mTOR in the Neurotoxicity of Rotenone-Induced Parkinson’s Disease Models

**DOI:** 10.3390/ijerph16203811

**Published:** 2019-10-10

**Authors:** Yu Zhang, Hui Guo, Xinyu Guo, Denfeng Ge, Yue Shi, Xiyu Lu, Jinli Lu, Juan Chen, Fei Ding, Qi Zhang

**Affiliations:** 1School of Medicine, Nantong University, 19 Qixiu Road, Nantong 226001, China; 1617062021@yxy.ntu.com (Y.Z.); Guoxinyu12580@163.com (X.G.); gdf1998@163.com (D.G.); 1617011042@yxy.ntu.edu.cn (Y.S.); yurixixi98@163.com (X.L.); 2Key Laboratory of Neuroregeneration of Jiangsu and Ministry of Education, Co-Innovation Center of Neuroregeneration, Nantong University, 19 Qixiu Road, Nantong 226001, China; 15896261344@163.com (H.G.); ljl602690814@163.com (J.L.); chenjuan0487561@icloud.com (J.C.); dingfei@ntu.edu.cn (F.D.); 3Jiangsu Clinical Medicine Center of Tissue Engineering and Nerve Injury Repair, 20 Xisi Road, Nantong 226001, China

**Keywords:** rotenone, Parkinson’s disease, neurotoxicity, Akt, mTOR

## Abstract

Rotenone has recently been widely used to establish Parkinson’s disease (PD) models to replicate the features of PD. However, the mechanisms involved in rotenone neurotoxicity have not been elucidated. The aim of the present study was to identify the neurotoxicity of rotenone through intraperitoneal injection in mice and to investigate the global changes of phosphorylation proteomic profiles in rotenone-injured SH-SY5Y cells through a label-free proteomic analysis using a PTMScan with LC–MS/MS. ICR (Institute of Cancer Research) mice were intraperitoneally injected with different dosages of rotenone (1 mg/kg/d or 3 mg/kg/d) daily for 21 consecutive days. Rotenone caused a dose-dependent decrease in locomotor activities and a decrease in the number of Nissl-positive and tyrosine hydroxylase (TH)-immunoreactive neurons in the substantia nigra pars compacta (SNpc). Here, 194 phosphopeptides on 174 proteins were detected in SH-SY5Y cells, and 37 phosphosites on 33 proteins displayed statistically significant changes in expression after rotenone injury. The downregulation of phosphorylated Akt and mTOR was further confirmed by western blot analysis. A specific Akt activator, SC79, could protect cell viability and induce autophagy in rotenone-injured SH-SY5Y cells. This study indicates the involvement of the Akt/mTOR (mammalian target of rapamycin) signaling pathway in rotenone-injured SH-SY5Y cells and provides molecular information for the neurotoxicity of rotenone.

## 1. Introduction

As a pesticide, rotenone has been widely used to establish Parkinson’s disease (PD) models in rodents and some nonmammalian models since it was first reported in 2000 [1,2]. Rotenone inhibits complex I of the mitochondrial electron transport chain, leading to reduced ATP (adenosine triphosphate) production and electron leakage. Both the systemic administration and regional injection of rotenone can reproduce the pathological hallmarks of PD as well as some Parkinsonian motor deficits, such as bradykinesia, rigidity, and a stooped-like posture [3,4]. In addition, studies with primary cultured dopaminergic neurons and SH-SY5Y cells have also shown the neurotoxicity of rotenone [5,6].

Several different routes of rotenone administration in rodents have been applied to establish PD models, including subcutaneous, intravenous, and intraperitoneal injection; stereotaxic infusion; inhalation; and intragastric administration [2]. As rotenone is highly lipophilic, it easily crosses all biological membranes, including the blood–brain barrier, which allows for the systemic administration of rotenone to develop animal models of PD [7]. It has been reported that the systemic administration of rotenone could induce chronic progressive degeneration of the nigrostriatal pathway and an α-synuclein pathology in rats, which is similar to what is seen in PD patients [1,8].

Although studies have not directly suggested that rotenone exposure is responsible for PD, exposure to environmental toxins increases the likelihood of the disease [5]. Chronic rotenone exposure might enhance oxidative and nitrosative stress, which induces mitochondrial dysfunction and apoptotic cell death in the striatum via the cytochrome c/caspase-3 signaling cascade [9]. Rotenone could also increase microglial proliferation and phagocytic activity, leading to neuronal loss [10]. A microarray analysis was employed to uncover pathogenic genes and the molecular pathological process in a cellular PD model induced by rotenone. There, 2174 differentially expressed genes were identified in the rotenone-induced PD model, and these genes were enriched in the pathways of cell cycle and protein processing in the endoplasmic reticulum [11]. However, the mechanisms involved in rotenone neurotoxicity were not elucidated.

Post-translational modifications (PTMs) are crucial mechanisms in cellular functions, as they regulate protein stability, localization, and function. PTMs have emerged as important modulators of α-synuclein and LRRK2 pathologies, which are important proteins in PD pathogenesis [6]. It has been reported that rotenone could induce α-synuclein phosphorylation via the phosphorylation of the catalytic subunit of PP2A (PP2Ac) at Tyr307 [12]. In nerve-like differentiated cells (NLCs), rotenone exposure significantly increased the p-LRRK2 kinase level, while the LRRK2 kinase inhibitor protected NLCs against rotenone-induced noxious effect [13]. In addition, the activation of the SIRT1/Akt1 signaling pathway decreased due to rotenone in PC12 cells [14]. The above evidence suggests that several proteins and cell signaling pathways could be affected by rotenone exposure. Further comprehensive investigation might be helpful in gaining insight into the neurotoxicity of rotenone in developing PD models.

In the present work, we aimed to investigate the neurotoxicity of rotenone through intraperitoneal injection in ICR (Institute of Cancer Research) mice and to detect the profile of protein phosphorylation in rotenone-injured SH-SY5Y cells through a comparative proteomic analysis: a PTMScan analysis. PTMScan Direct is an immunoaffinity-based LC–MS/MS method for the analysis of important signaling proteins, and it enables the identification and quantification of hundreds of peptides derived from specific proteins in signaling pathways or specific protein types. The PTMScan Direct Multi-Pathway reagent was used to study the mechanisms of rotenone injury in human neuroblastoma SH-SY5Y cells. Our findings provide a new understanding of the molecular mechanisms involved in rotenone-induced neuronal injury and the potential therapeutic targets for PD treatment.

## 2. Experimental Procedure

### 2.1. Chemicals and Antibodies

Rotenone, trypsin, and monoclonal mouse anti-β-actin antibody were purchased from Sigma Aldrich (St. Louis, MO, USA). Rabbit antityrosine hydroxylase (anti-TH) antibody, rabbit anti-p-Akt (Thr308) antibody, mouse anti-Akt antibody, rabbit anti-p-mTOR (Ser2448) antibody, rabbit anti-mTOR antibody, and rabbit anti-Atg5 antibody were from Abcam (Cambridge, MA, USA). Monoclonal anti-LC3 mAb-HRP-DirecT was from MBL (Woburn, MA, USA). Dulbecco’s Modified Eagle’s Medium (DMEM) and fetal bovine serum (FBS) were purchased from Gibco (Grand Island, NY, USA). Cell lysis buffer, protease inhibitor cocktail, a BCA (bicinchoninic acid) -based protein quantification kit, and BeyoECL Plus were obtained from Beyotime (Shanghai, China). A PTMScan^®^ Multi-Pathway Enrichment Kit, HRP (horse radish peroxidase)-conjugated goat anti-mouse IgG, and HRP-conjugated donkey anti-rabbit IgG were from Cell Signaling Technology (Danvers, MA, USA).

### 2.2. Animals

Adult ICR mice (25–30 g) were obtained from the Experimental Animal Center of Nantong University (China). All experiments involving animals were conducted in accordance with the institutional animal care guidelines of Nantong University.

Different dosages of rotenone [1 mg/kg/d or 3 mg/kg/d, dissolved in normal saline with 2% DMSO (dimethyl sulfoxide)] were applied to ICR mice through intraperitoneal injection for 21 consecutive days to establish a PD model. Normal saline with 2% DMSO was used as the control.

### 2.3. Behavioral Tests

At the end of the injection, the mice were subjected to training involving a rotarod test, pole test, and traction test for 3 days before behavioral tests were performed. The rotarod test was used to evaluate the motor coordination in PD mice using an accelerating rotarod apparatus (RWD, Shenzhen, China), as described previously [15]. The mice were trained for 3 consecutive days after the last injection in an acceleration mode (4–40 rpm) for 5 min. For the test, mice were acclimated to the testing room for at least 30 min and then placed on a rotating rod, which was accelerated from 4 to 40 rpm over a 5-min period. The latency before each mouse fell off the rod was recorded. The pole test was conducted to reflect bradykinesia, with minimal modifications (as described previously) [15]. Briefly, mice were placed head-upward at the top of a rough-surfaced wooden pole (16 mm in diameter and 60 mm in height). The total time for each mouse to orient downward and descend to the floor was recorded. Three examinations were conducted for each mouse, and the average values were calculated. The traction test was conducted as follows: the mice’s forepaws were suspended from a horizontal wire about 70 cm off of the ground. Scoring was divided from 1 to 3 points: 3 points = the mouse grasped the wire with two hind limbs; 2 points = the mouse grasped the wire with one hind limb; 1 point = the mouse could not grasp the wire with either hind limb [16]. Ten animals in each group were used for the behavioral tests.

### 2.4. Nissl Staining

Nissl staining was performed to observe the neuronal loss in the substantia nigra pars compacta (SNpc). After behavioral tests, mice were sacrificed by transcardial perfusion sequentially with saline and 4% paraformaldehyde in 0.1 M PBS, and the midbrain tissues were dissected and fixed. The specimens were washed in water, dehydrated in a graded ethanol series, cleared in xylene, embedded in paraffin, and cut into 5-μm-thick sections. Nissl staining was applied to the sections before photographs were taken with a microscope (Leica, Heidelberg, Germany). Three animals in each group were used for the experiments. The integrated densities (IntDen) of Nissl staining in SNpc were quantified with Image J software (version 1.51, NIH, Bethesda, MD, USA).

### 2.5. TH Immunofluorescent Staining

The midbrain sections were incubated with anti-TH antibody (1:200) overnight at 4 °C and donkey anti-rabbit IgG-FITC (fluorescein) (1:200) for 2 h at room temperature. The total number of TH-positive cells was counted in sections using an unbiased stereological method, as previously described [17]. Three animals in each group were used for the experiments.

### 2.6. Cell Culture and Treatment

Human SH-SY5Y neuroblastoma cells were cultured in DMEM supplemented with 10% FBS at 37 °C in a humidified atmosphere of 95% air and 5% CO_2_. The day before treatment, cells were plated in a 10-cm culture dish at a density of 1 × 10^6^ cells/mL. The cells were incubated with or without 100 μM of rotenone for 30 min and then washed once with cold PBS and scraped in urea lysis buffer (9 M of sequanol-grade urea, 20 mM HEPES (N-2-hydroxyethylpiperazine-N-ethane-sulphonicacid), pH 8.0, 1 mM β-glycerophosphate, 1 mM sodium vanadate, 2.5 mM sodium pyro-phosphate). For the cell viability measurement and transmission electron microscopy observations, the cells were treated with 100 μM rotenone for 24 h. In addition, 8 μg/mL SC79 was applied to activate the Akt signal pathway.

### 2.7. PTMScan Multipathway Analysis

The cells in lysis buffer were sonicated three times for 20 s each with 1 min of cooling on ice between each burst. Sonicated lysates were centrifuged for 15 min at 4 °C at 20,000× *g*. An aliquot of each supernatant was reserved for western blotting and stored at −80 °C. Supernatants were collected and reduced with 4.5 mM DTT (dithiothreitol) for 30 min at 55 °C. Reduced lysates were alkylated with iodoacetamide (0.019 g/mL) for 15 min at room temperature in the dark. The samples were diluted 1:4 with 20 mM HEPES, pH 8.0, and were digested overnight with 10 g/mL trypsin-tosylphenylalanyl chloromethyl ketone (Worthington, Lakewood, NJ, USA) in 1 mM HCl. Digested peptide lysates were acidified with 1% TFA (tallow fatty acid), and peptides were desalted over 360 mg of SEP PAK Classic C18 columns (Waters, Milford, MA, USA). Peptides were eluted with 40% acetonitrile in 0.1% TFA, dried under vacuum, and stored at −80 °C. The PTMScan Direct analysis was adapted from the PhosphoScan method developed at Cell Signaling Technology [18].

### 2.8. Western Blot Analysis

Protein concentration was quantified by BCA analysis, followed by electrophoresis separation on SDS-PAGE. After transfer to a PVDF (polyvinylidene fluoride) membrane (Millipore, Bedford, MA, USA), the membrane was blocked with 5% nonfat dry milk in Tris-buffered saline (TBS, pH 7.4) and incubated with anti-p-Akt (1:500), anti-Akt (1:500), anti-p-mTOR (1:500), anti- mTOR (1:500), anti-Atg5 (1:500), anti-LC3B (1:500), or anti-β-actin (1:4000) at 4 °C overnight. After a wash with TBS/T (TBS with 0.1% Tween 20), HRP-conjugated affinity-purified donkey anti-rabbit IgG (1:5000) or goat anti-mouse IgG (1:5000) was applied at room temperature for 1 h. The membrane was incubated with an ECL (electrogenerated chemiluminescence) substrate and scanned with an imaging system (Tanon, Shanghai, China). The data were analyzed with PDQuest 7.2.0 software (Bio-Rad, Hercules, CA, USA), and β-actin was used as an internal control protein for normalizing protein loading.

### 2.9. Cell Viability Measurement

The cell viability was assessed through the CCK-8 method. Briefly, tetrazolium salt-8 (WST-8) solution was added to cultured cells in 96-well plates (10 μL/each well). After incubation at 37 °C for 1 h, the absorbance (optical density, OD) was measured by spectrophotometry at 450 nm with an ELx-800 microplate reader (Bio-Tek Inc., Winooski, VT, USA).

### 2.10. Transmission Electron Microscopy (TEM)

After treatment, the SH-SY5Y cells were fixed with precooled 2.5% glutaraldehyde, postfixed with 1% osmium tetraoxide solution, dehydrated stepwise in increasing concentrations of ethanol, and embedded in Epon 812 epoxy resin. Ultrathin sections were stained with lead citrate and uranyl acetate and then examined under a transmission electron microscope (JEOL Ltd., Tokyo, Japan).

### 2.11. Statistical Analysis

Data are expressed as mean ± SEM (standard error of mean). Statistical analyses were performed with GraphPad Prism 6.0 (GraphPad Software Inc., San Diego, CA, USA). One-way analysis of variance (ANOVA) was used to identify significant differences between experimental groups, and a subsequent Bartlett’s multiple comparisons test was used to compare the differences between each group: *p* < 0.05 was considered statistically significant.

## 3. Results

### 3.1. Rotenone-Reduced Motor Functions of Mice

Different concentrations of rotenone were injected intraperitoneally for 21 consecutive days to establish a mouse PD model. Motor functions were assessed through a rotarod test, pole test, and traction test at the end of injection. In the rotarod test, a significant shorter latency in falling was observed in the high-dose rotenone group (3 mg/kg/d) compared to the control group (*p* < 0.05) (Figure 1A). The time spent in descending from the pole increased (*p* < 0.05), while the score of the traction test decreased (*p* < 0.01) in the 3-mg/kg/d rotenone group (Figure 1B,C). Although there was no significant difference between the low-dose rotenone group (1 mg/kg/d) and the control group in any behavioral test, the above data showed a tendency to decrease or increase with increasing doses, suggesting that the intraperitoneal injection of rotenone could reduce the motor functions of ICR mice.

### 3.2. Rotenone-Induced Neuronal Loss in SNpc

To observe the effects of rotenone on neuronal loss in the SNpc, Nissl staining was performed, and the images are shown in Figure 2A. Compared to the control group, the cell density in the SNpc decreased in the 1-mg/kg/d rotenone group (*p* < 0.05) and the 3-mg/kg/d rotenone group (*p* < 0.01) (Figure 2B). Additionally, we detected TH-positive neurons in the SNpc after rotenone injection. The images showed that both a low dose and high dose of rotenone could decrease TH-positive cells in the SNpc (*p* < 0.05 or *p* < 0.01), which was consistent with the Nissl staining analysis (Figure 2C,D). These data suggest that rotenone could induce neuronal loss in the SNpc, which might have been responsible for the deficits in motor functions observed in the behavioral tests.

### 3.3. Global Changes in Phosphorylation Proteomic Profiles in Rotenone-Injured SH-SY5Y Cells

To further investigate the signal pathway involved in rotenone neurotoxicity, a rotenone-induced SH-SY5Y cell model was applied. Our previous studies proved that rotenone could induce mitochondrial dysfunction and cell apoptosis in both SH-SY5Y cells and midbrain dopaminergic neurons [17]. In this study, SH-SY5Y cells were treated with 100 μM rotenone for 30 min and then harvested, digested, and immunoprecipitated with PTMScan Direct reagents. Overall, 194 phosphopeptides on 174 proteins were detected in both groups, with 44 peptides on 33 proteins only detected in the rotenone group and 44 peptides on 42 proteins only detected in the control group (Figure 3A). Compared to the control cells, 37 phosphosites on 33 proteins displayed statistically significant changes in expression (±2-fold, *p* < 0.05) in the rotenone group, in which 10 phosphosites were upregulated and 27 phosphosites were downregulated after rotenone treatment. A full list of the phosphoproteins identified that were differentially expressed between the two groups is available in Table 1. A gene ontology (GO) analysis revealed that these proteins were mainly involved in the biological processes (BPs) of epithelial cell differentiation and the regulation of protein phosphorylation. The molecular functions (MFs) related to protein heterodimerization activity, ubiquitin protein ligase binding, mRNA binding, and protein dimerization activity were also affected by rotenone injury (Figure 3B,C). To further explore the impact of differentially expressed phosphoproteins in the cell physiological process and to discover the internal relations between differentially expressed proteins, a Kyoto Encyclopedia of Genes and Genomes (KEGG) enrichment analysis was performed, and the data indicated that the most significantly affected pathway due to rotenone treatment was apoptosis. A protein–protein interaction (PPI) network was generated (Figure 3D). The network consisted of a complex of interconnected proteins with a number of differentially expressed phosphoproteins. Among these signal pathways, the MAPK8 (mitogen-activated protein kinase 8, also known as JNK1)/c-JUN pathway, which plays a critical role in the apoptosis process, was activated by rotenone, while the PI_3_K/Akt/mTOR pathway, which is an intracellular signaling pathway important in regulating the cell cycle and autophagy, was inactivated in rotenone-injured cells.

### 3.4. The Phosphorylation of Akt/mTOR Signaling Decreased in Rotenone-Injured SH-SY5Y Cells

Among the differentially expressed phosphorylated peptides in Table 1, we found that phosphorylated threonine at position 165 (Thr165) of the Akt peptide fragment was not detected after rotenone treatment. At the same time, the phosphorylated serine at position 104 (Ser104) of the mTOR peptide fragment decreased to about 20% in rotenone-injured cells compared to the control. Western blot was used to validate the results of the proteomic approach. The expressions of phosphorylated Akt (Thr308) and mTOR (Ser2448) decreased with 100 μM of rotenone treatment over 30 min (*p* < 0.01), while the total Akt and mTOR did not significantly change (Figure 4A–C). The two phosphorylation sites recognized by the antibodies were the same positions detected by the PTMScan method. These data confirmed that the Akt/mTOR pathway was inactivated by rotenone treatment.

### 3.5. The Activation of Akt/mTOR Reduced the Neurotoxicity of Rotenone and Induced Autophagy

SC79 is a specific Akt activator that enhances Akt phosphorylation and activation in multiple cell types. We first demonstrated by western blot that SC79 at 8 μg/mL significantly promoted Akt and mTOR phosphorylation in SH-SY5Y cells (*p* < 0.05 or *p* < 0.01) (Figure 4A–C). The decreased cell viability by rotenone could be rescued by SC79 treatment (*p* < 0.01) (Figure 5A), suggesting that the activation of Akt/mTOR was able to reduce the neurotoxicity of rotenone. Our previous research indicated that mitochondrial dysfunction was responsible for rotenone neurotoxicity. Here, we further confirm that rotenone induced morphological changes in the mitochondria, including mitochondrial swelling, mitochondrial crest fracture, and mitochondrial vacuolar degeneration in SH-SY5Y cells (Figure 5B). SC79 treatment significantly protected the cells from the above pathological changes and induced the formation of autophagic vacuoles to clear the degenerated mitochondria (Figure 5B). In addition, the expression of autophagy-related protein 5 (Atg5) and the ratio of LC3B-II to LC3B-I were upregulated by SC79 treatment (*p* < 0.01) (Figure 4D–F). Together, these results demonstrate that the activation of the Akt/mTOR signaling pathway might induce autophagy and protect SH-SY5Y cells from rotenone injury.

## 4. Discussion

Rotenone-induced PD mice models have been frequently used in recent studies to investigate the mechanisms of PD. However, limited data are available currently concerning the molecular mechanisms of the pathological changes induced by rotenone. In this study, we explored the effects of rotenone in vivo through intraperitoneal injection and further investigated the profile of protein phosphorylation in vitro through PTMscan Direct combined with a comparative label-free LC–MS/MS technique to provide information about the mechanisms involved in rotenone neurotoxicity.

Our results confirmed that the intraperitoneal injection of rotenone reproduced the characteristic behavioral and histopathological features of PD, including a dose-dependent decrease in motor functions and a loss of neuronal cells in SNpc.

Previously, several studies demonstrated that the expression of some important proteins, as well as the activities of certain enzymes, significantly changed after rotenone injury (immunoblotting analyses or biochemical tests) [19,20]. In the present study, a multipathway proteomic method was applied to expand the list of proteins playing specific roles in response to rotenone. PTMscan Direct is an antibody-based proteomic method for the identification and quantification of post-translationally modified peptides, which have been employed to study signaling pathways involved in neurodegenerative disease [18,21]. This method allows for the quantification of hundreds of phosphorylated peptides and proteins in many different cell signaling pathways, including Akt signaling, MAP kinase signaling, and apoptosis. As protein phosphorylation plays an essential role in the regulation of cellular processes, exploring global changes in phosphorylation proteomic profiles might be helpful in completing our understanding of rotenone injury and further disclosing the pathology of PD.

Protein phosphorylation is usually an instant change in response to a stimulus. In this study, the establishment of a mouse PD model with rotenone injection was a long-term process. As an earlier response of the cells to rotenone injury was required to further understand the mechanisms involved in its neurotoxicity, mouse brain tissue harvested after 3 weeks of rotenone injection was not appropriate for detection of the phosphorylation changes after injury. The human neuroblastoma SH-SY5Y cell line is a commonly used cell line in studies related to neurodegenerative diseases. A systems genomics approach has shown that most of the genes belonging to the major PD pathways and modules are intact in the SH-SY5Y genome. Specifically, each analyzed gene related to PD has at least one intact copy in SH-SY5Y cells [22]. Thus, we chose SH-SY5Y cells to establish a PD model in vitro and to do a proteomic analysis. By using a multipathway reagent in the PTMScan Direct method, we detected 194 phosphopeptides on 174 proteins in SH-SY5Y cells. The GO analysis and KEGG analysis indicated that apoptosis was the most significantly affected process with rotenone treatment. The PPI analysis indicated that the PI_3_K/Akt/mTOR pathway was inactivated in rotenone-injured cells. In addition, the western blot confirmed the differences in the expressions of phosphorylated Akt (Thr308) and mTOR (Ser2448), which were consistent with the PTMScan data. PI_3_K/Akt is usually activated by receptors of neurotrophins and growth factors, which is important in the regulation of cell growth, differentiation, survival, apoptosis, and autophagy [23]. Dysregulation of the PI_3_K/Akt/mTOR pathway, such as decreased phosphorylated Akt and suppressed mTOR, is commonly reported in brains and dopaminergic neurons from PD patients and contributes to the loss of dopaminergic neurons in PD [24,25,26]. Our data indicated that the inactivation of the Akt/mTOR pathway might be involved in the neurotoxicity of rotenone.

To further detect the role of the Akt/mTOR pathway in the rotenone-injured cells, SC79, a small molecule that inhibits Akt membrane translocation and activates Akt phosphorylation in the cytosol, was applied. SC79 could protect DA neuronal cells from mitochondrial toxins, possibly via the activation of Akt/Nrf2 signaling [27]. It has also been reported that SC79 protects retinal pigment epithelium cells from UV radiation via activation of the Akt/Nrf2 signaling pathway [28]. Here, we found that SC79 activated the Akt/mTOR signaling pathway and inhibited cell death induced by rotenone, further confirming the requirement of inactivation of Akt/mTOR in rotenone neurotoxicity in vitro. However, the in vivo protective effects of the Akt activator should be further tested. Although the cytosolic activation of Akt by SC79 was sufficient to recapitulate the primary cellular function of Akt signaling, which protected ischemic-injury-induced neuronal death in a hippocampal neuronal culture system and a mouse model for ischemic stroke [29], this was a short-term application in vivo in an ischemic stroke model. Some research has also found that the activation of the PI3K/Akt signaling pathway could protect neuronal cells from apoptosis induced by heat shock and UV light in zebrafish [30]. However, in the case of a PD mice model, long-term drug delivery should be considered, and efforts should be made to find some natural compounds, such as resveratrol [14], that have the ability to activate Akt in vivo and demonstrate the protective effects of these compounds in mice PD models.

Recent research has indicated that autophagy was also involved in the pathogenesis of rotenone-induced PD. The injection of rotenone into rats’ brains could induce PD pathology as well as increase autophagosome vesicles [31]. Autophagosome vesicles were also observed in the brain tissue of PD patients, indicating a disruption of autophagic flux [32]. However, the functions of autophagy in neurodegenerative diseases remain controversial. Some research has reported that inducible autophagy contributed to survival in stress conditions [25,32]. The autophagic process was altered in neurodegenerative diseases and modulated by PI_3_K/Akt/mTOR activity, while the interplay between them was complicated [23,24]. As a specific marker of autophagy, the LC3-II/I ratio has frequently been used to examine the change in autophagic activity [33], and Atg5 is a key protein that is necessary for LC3 conjugation to phosphatidylethanolamine to form LC3-II [34,35]. In this study, although we observed some autophagic vacuole formation through TEM in SH-SY5Y cells with rotenone injury, no significant change in expression in the Atg5 or LC3B-II/I ratio was detected. In addition, the activation of Akt/mTOR induced the formation of autophagic vacuoles and increased Atg5 expression and the LC3B-II/I ratio. These data indicate that SC79 might protect rotenone-injured cells through the induction of autophagy, which is responsible for the clearance of dysfunctional mitochondria and protein aggregates in neurodegeneration [24].

## 5. Conclusions

The current study demonstrates that the systematic administration of rotenone affected motor functions and induced neuronal loss in mice SNpc. The inactivation of the Akt/mTOR signaling pathway was involved in rotenone-injured SH-SY5Y cells, and the activation of Akt/mTOR might have induced autophagy to exert protective effects. However, other mechanisms for the neurotoxicity of rotenone in PD pathology should be further investigated.

## Figures and Tables

**Figure 1 ijerph-16-03811-f001:**
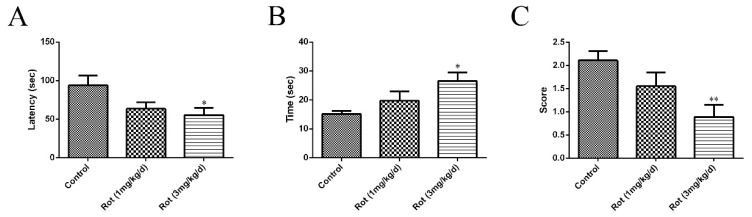
Rotenone caused dose-dependent changes in behavioral tests in mice: 1 mg/kg/d or 3 mg/kg/d of rotenone (dissolved in normal saline with 2% DMSO) was applied to ICR mice through intraperitoneal injection for 21 consecutive days to establish a Parkinson’s disease (PD) model. Normal saline with 2% DMSO was used in the control group. (**A**) The time of latency in falling was recorded after the last injection of rotenone. (**B**) The total time for each mouse to orient downward and descend to the floor was recorded. (**C**) The score of the traction test was assessed in each group. Data are represented as the mean ± SEM (*n* = 10); * *p* < 0.05 and ** *p* < 0.01 versus control group.

**Figure 2 ijerph-16-03811-f002:**
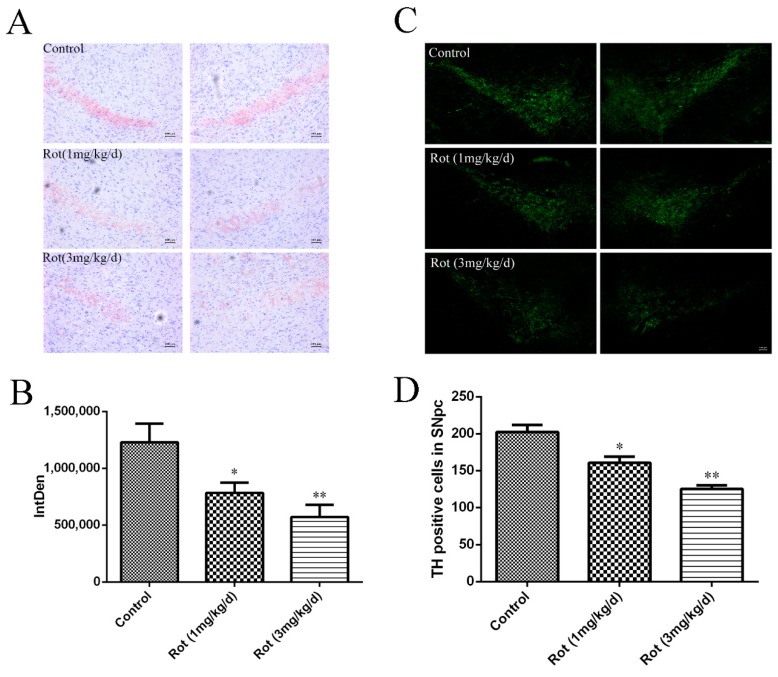
Rotenone induced neuronal loss in the substantia nigra pars compacta (SNpc) of mice. (**A**) Nissl staining of the SNpc in mice with different doses of rotenone injection. Scale bar = 100 μm. (**B**) Quantitative analysis of Nissl staining in the SNpc (*n* = 3). (**C**) Tyrosine hydroxylase (TH) immunostaining (green) of the SNpc in a rotenone-induced PD mice model. Scale bar = 100 μm. (**D**) Quantitative analysis of TH immunoreactivity in the SNpc. * *p* < 0.05 and ** *p* < 0.01 versus control group (*n* = 3).

**Figure 3 ijerph-16-03811-f003:**
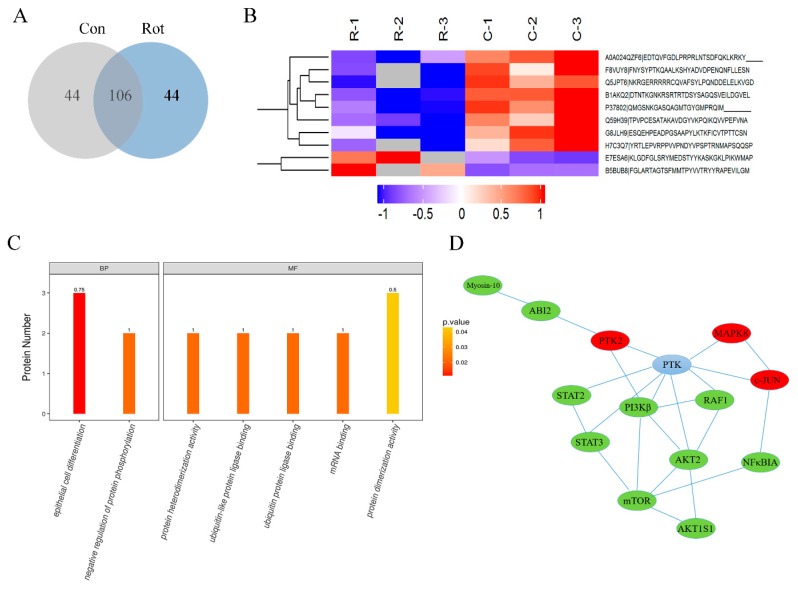
Proteomic profiling of phosphorylation in rotenone-injured SH-SY5Y cells. SH-SY5Y cells were incubated with or without 100 μM of rotenone for 30 min and then harvested for PTMScan multipathway analysis. (**A**) Venn diagram to show the number of overlapping phosphopeptide identifications between the rotenone group (Rot) and control group (Con). (**B**) Hierarchical cluster of phosphopeptides differentially expressed between the rotenone group and control group. Each row represents a different phosphopeptide. Each column represents a different treatment, as indicated. C-1, C-2, and C-3 represent three replicates for the control group. R-1, R-2, and R-3 represent three replicates for the rotenone group. Red means an increase in abundance relative to the control. Blue means a decrease in abundance relative to the control. (**C**) Gene ontology (GO) enrichment analyses of phosphorylated proteins. The proteins were classified as biological processes (BPs) and molecular functions (MFs). (**D**) Protein–protein interaction (PPI) network analysis of differentially expressed phosphoproteins. Upregulated proteins are highlighted in red, and downregulated proteins are highlighted in green. PTK = protein tyrosine kinase, PTK2 = focal adhesion kinase 1, ABI2 = Abl interactor 2, MAPK8 = mitogen-activated protein kinase 8, RAF1 = RAF proto-oncogene serine/threonine-protein kinase, PI3Kβ = phosphatidylinositol-4,5-bisphosphate 3-kinase catalytic subunit beta isoform, NF κ BIA = NF-kappa-B inhibitor alpha, Akt = RAC-beta serine/threonine-protein kinase, mTOR = mammalian target of rapamycin, STAT2/3 = signal transducer and activator of transcription 2/3, Akt1S1 = Akt1 substrate 1.

**Figure 4 ijerph-16-03811-f004:**
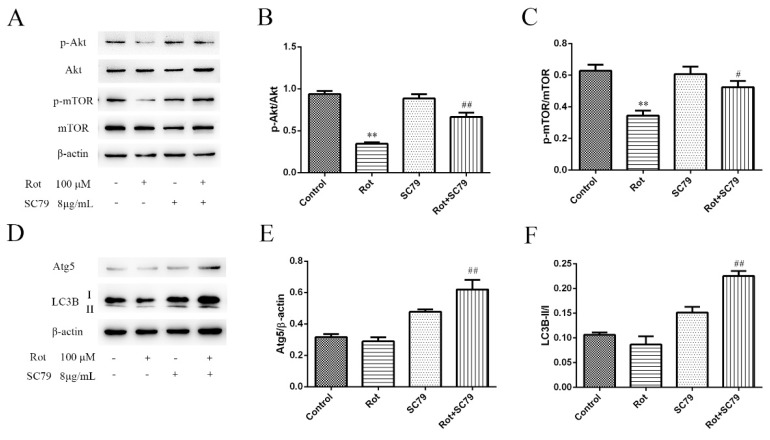
Western bolt validation of inactivated Akt/mTOR signaling and autophagy-related proteins in rotenone-injured SH-SY5Y cells. (**A**) SH-SY5Y cells were incubated with rotenone for 30 min, and then both phosphorylated and total Akt and mTOR were detected by western blot analysis. Here, 8 μg/mL SC79 was applied to activate the Akt pathway, and β-actin served as the internal control. (**B**,**C**) A quantification of phosphorylated protein expressions, measured in Panel A. Data are expressed as the ratio of phosphorylated proteins to total proteins. ** *p* < 0.01 versus control group; ^#^
*p* < 0.05, ^##^
*p* < 0.01 versus rotenone group (*n* = 3). (**D**) The protein expression of Atg5 and LC-3B in rotenone-injured SH-SY5Y cells was examined by western blot. (**E**) A quantification of Atg5 expression. The data are expressed as the ratio of Atg5 to β-actin. ^##^
*p* < 0.01 versus rotenone group (*n* = 3). (**F**) A quantification of LC3B expression. The data are expressed as the ratio of LC3B-II to LC3B-I. ^##^
*p* < 0.01 versus rotenone group (*n* = 3).

**Figure 5 ijerph-16-03811-f005:**
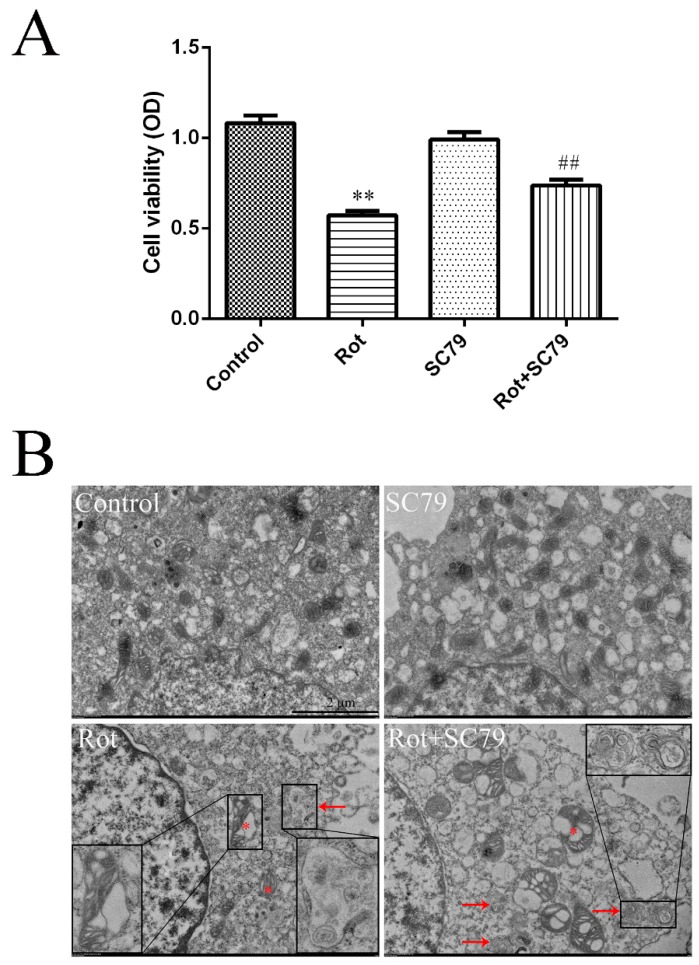
The activation of Akt/mTOR by SC79 protected SH-SY5Y cells from rotenone injury. SH-SY5Y cells were incubated with rotenone and SC79 for 24 h. (**A**) Cell viability was measured through a CCK-8 (cell counting kit-8) assay, and the absorbance (optical density, OD) was measured by spectrophotometry at 450 nm. ** *p* < 0.01 versus control group; ^##^
*p* < 0.01 versus rotenone group (*n* = 18). (**B**) Representative TEM images of SH-SY5Y cells treated with rotenone and SC79. Asterisks indicate degenerated mitochondria. Arrows indicate autophagic vacuoles. Scale bar = 2 μm.

**Table 1 ijerph-16-03811-t001:** Differentially expressed phosphoproteins between the rotenone group and control group.

Protein Name	Gene Name	Position	Amino Acid	Sequence Window	Charge	Average R ^a^	Average C ^b^	R/C ^c^
Mitogen-activated protein kinase (fragment)	*MAPK8*	185	Y	FGLARTAGTSFMMTPYVVTRYYRAPEVILGM	2	899,695,000	60,972,333	14.76
Focal adhesion kinase 1	*PTK2*	620	Y	KLGDFGLSRYMEDSTYYKASKGKLPIKWMAP	2	32,010,500	11,210,166	2.86
Transcription factor AP-1	*JUN*	73	S	SDLLTSPDVGLLKLASPELERLIIQSSNGHI	3	218,540,000	75,921,666	2.88
Akt1 substrate 1 (proline-rich), isoform CRA_a	*Akt1S1*	246	T	EDTQVFGDLPRPRLNTSDFQKLKRKY	2	996,283,333	2,161,533,333	0.46
Signal transducer and activator of transcription	*STAT3*	607	Y	ESQEHPEADPGSAAPYLKTKFICVTPTTCSN	3	95,216,000	231,736,666	0.41
Abl interactor 2 (fragment)	*ABI2*	79	Y	YRTLEPVRPPVVPNDYVPSPTRNMAPSQQSP	3	21,101,000	53,364,000	0.40
GIG10		108	S	NKRGERRRRRCQVAFSYLPQNDDELELKVGD	2	70,922,000	202,776,666	0.35
Signal transducer and activator of transcription (fragment)		700	Y	TPVPCESATAKAVDGYVKPQIKQVVPEFVNA	2	83,187,000	238,250,000	0.35
Sodium-coupled neutral amino acid transporter 2	*SLC38A2*	41	Y	FNYSYPTKQAALKSHYADVDPENQNFLLESN	2	15,504,000	45,813,333	0.34
Transgelin-2	*TAGLN2*	192	Y	QMGSNKGASQAGMTGYGMPRQIM	2	22,752,666	68,413,666	0.33
Serine/threonine-protein kinase mTOR (fragment)	*MTOR*	104	S	DTNTKGNKRSRTRTDSYSAGQSVEILDGVEL	2	35,380,000	176,786,666	0.20
Tubulin alpha-1B chain	*TUBA1B*	223	T	AIYDICRRNLDIERPTYTNLNRLIGQIVSSI	3	15,692,500	— ^d^	
La-related protein 1	*LARP1*	631	S	MDGRKNTFTAWSDEESDYEIDDRDVNKILIV	3	21,394,333	—	
La-related protein 1	*LARP1*	627	S	EMEQMDGRKNTFTAWSDEESDYEIDDRDVNK	3	21,394,333	—	
cDNA FLJ60109, highly similar to RUN and SH3 domain-containing protein 2		543	S	PAAMAGPGSPPRRVTSFAELAKGRKKTGGSG	2	21,557,500	—	
cDNA FLJ40872 fis, clone TUTER2000283, highly similar to *Homo sapiens* transformer-2-beta (SFRS10) gene		230	S	DRYEDYDYRYRRRSPSPYYSRYRSRSRSRSY	3	64,762,500	—	
cDNA FLJ40872 fis, clone TUTER2000283, highly similar to *Homo sapiens* transformer-2-beta (SFRS10) gene		228	S	GWRAAQDRDQIYRRRSPSPYYSRGGYRSRSR	3	64,762,500	—	
Rho-related GTP-binding protein RhoC (fragment)	*RHOC*	34	Y	TCLLIVFSKDQFPEVYVPTVFENYIADIEVD	2	90,055,500	—	
ADAM metallopeptidase domain 10, isoform CRA_b	*ADAM10*	740	S	PPQPIQQPQRQRPRESYQMGHMRR	4	—	11,429,000	
Tyrosine-protein kinase		32	Y	VDLKTQPVRNTERTIYVRDPTSNKQQRPVPE	2	—	11,667,133	
Serine/threonine-protein kinase mTOR (fragment)	*MTOR*	110	S	NKRSRTRTDSYSAGQSVEILDGVELGEPAHK	4	—	46,053,666	
Histone H2B	*HIST1H2BI*	65	S	HPDTGISSKAMGIMNSFVNDIFERIAGEASR	2	—	180,860,000	
Kin of IRRE-like protein 1	*KIRREL*	637	S	EAYDPIGKYATATRFSYTSQHSDYGQRFQQR	3	—	21,853,500	
cDNA FLJ51708, highly similar to phosphatidylinositol-4,5-bisphosphate 3-kinase catalytic subunit beta isoform		29	Y	KVKTKKSTKTINPSKYQTIRKAGKVHYPVAW	2	—	17,646,400	
cDNA FLJ58463, highly similar to myosin-10		512	S	TKTFTPCERLEKRRTSFLEGTLRRSFRTGSV	3	—	42,488,333	
cDNA FLJ50355, highly similar to RAF proto-oncogene serine/threonine-protein kinase		123	S	GTQEKNKIRPRGQRDSSYYWEIEASEVMLST	3	—	68,606,333	
Tripartite motif-containing 3, isoform CRA_f (fragment)	*TRIM3*	308	S	SPFRVRALRPGDLPPSPDDVKRRVKSPGGPG	3	—	8,615,450	
NF-kappa-B inhibitor alpha	*NFKBIA*	32	S	RDGLKKERLLDDRHDSGLDSMKDEEYEQMVK	3	—	14,951,000	
AP complex subunit beta	*AP2B1*	276	Y	KVLMKFLELLPKDSDYYNMLLKKLAPPLVTL	2	—	36,866,666	
RAC-beta serine/threonine-protein kinase (fragment)	*AKT2*	165	T	FGLCKEGISDGATMKTFCGTPEYLAPEVLED	3	—	256,295,000	
ARF GTPase-activating protein GIT1	*GIT1*	545	Y	RLQPFHSTELEDDAIYSVHVPAGLYRIRKGV	3	—	46,371,500	
Aurora kinase C (fragment)	*AURKC*	26	T	SEKLDEQRTATVRRKTMCGTLDYLPPEMIEG	2	—	148,152,333	
Serine/threonine-protein kinase N1	*PKN1*	914	T	TDVSNFDEEFTGEAPTLSPPRDARPLTAAEQ	3	—	965,445,000	
MYO1E variant protein	*MYO1E*	989	Y	YPHAPGSQRSNQKSLYTSMARPPLPRQQSTS	3	—	7,483,433	
Enhancer of mRNA-decapping protein 3	*EDC3*	150	S	QQCSKSYVDRHMESLSQSKSFRRRHNSWSSS	2	—	11,110,500	
PCDHGC3 protein		24	Y	PQFTLQHVPDYRQNVYIPGSNATLTNAAGKR	3	—	19,835,000	
Ubiquinol-cytochrome-c reductase complex assembly factor 2 (fragment)	*UQCC2*	67	Y	ACDQMYESLARLHSNYYKHKYPRPRDTSFSG	3	—	6,781,933	

^a^ Average R = average intensity of phosphorylated signal in three rotenone groups; ^b^ average C = average intensity of phosphorylated signal in three control groups; ^c^ R/C = the ratio of average R/average C; ^d^ — = not detectable.

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
