# Peer review of "Involvement of Akt/mTOR in the Neurotoxicity of Rotenone-Induced Parkinson’s Disease Models"

_ijerph, 2019, doi:10.3390/ijerph16203811_

Round 1
Reviewer 1 Report
Authors demonstrated that Rotenone are able to induce PD features, however, they use a cell linage (SH-SY5Y cells) to do the proteomic analysis. Authors should discuss why they use cell linage instead of primary cells obtained from the mouse model used. Why SH-SY5Y cells are the best choice and how this is applicable for PD disease.Authors validate their proteomic results, which is quite impressive. Authors also suggest that Akt/mTOR pathway activation may protects neurotoxicity, however, they do not demonstrated in vivo. Authors should demonstrated in the mouse model that the activation of Akt/mTOR pathway ameliorates motor functions and neuronal loss in SNpc of mice after exposure to rotenone, thus then they can suggest that the pathway can be a potential therapeutic target.
Author Response
Reviewer 1
Authors demonstrated that Rotenone are able to induce PD features, however, they use a cell linage (SH-SY5Y cells) to do the proteomic analysis. Authors should discuss why they use cell linage instead of primary cells obtained from the mouse model used. Why SH-SY5Y cells are the best choice and how this is applicable for PD disease.
Response: Thank you for the reviewing. For the proteomic analysis, 15 dishes of cells growing in 10-cm dish were required for each group at least. We did try to harvest primary mouse midbrain cells. However, the number of cells was far from the requirement. The human neuroblastoma cell line, SH-SY5Y, is a commonly used cell line in studies related to neurodegenerative diseases. Systems genomics approach showed that most of the genes belonging to the major Parkinson’s disease pathways and modules were intact in the SH-SY5Y genome. Specifically, each analyzed gene related to PD had at least one intact copy in SH-SY5Y cells (Systems genomics evaluation of the SH-SY5Y neuroblastoma cell line as a model for Parkinson’s disease. Krishna et al. BMC Genomics 2014, 15:1154). Thus, we chose SH-SY5Y cells to establish a PD model in vitro and to do the proteomic analysis.
Authors validate their proteomic results, which is quite impressive. Authors also suggest that Akt/mTOR pathway activation may protects neurotoxicity, however, they do not demonstrated in vivo. Authors should demonstrated in the mouse model that the activation of Akt/mTOR pathway ameliorates motor functions and neuronal loss in SNpc of mice after exposure to rotenone, thus then they can suggest that the pathway can be a potential therapeutic target.
Response: Thank you for the advice. It would be better to confirm the protective effects of Akt activator in vivo. Identifying effective Akt activator is in great need for a variety of therapeutic applications and rare Akt activator was ultimately successful. In this study, we used SC79, a small molecule that inhibits Akt membrane translocation and activates Akt in the cytosol, to activate Akt signaling in vitro. It was reported that SC79 could enhance Akt activity during neuronal cell death in an in vivo mouse ischemia model (Small molecule-induced cytosolic activation of protein kinase Akt rescues ischemia-elicited neuronal death. Jo et al. Proc Natl Acad Sci U S A. 2012, 109(26):10581-6). However, it was a short-term application in vivo in above study. In the case of PD, long-term drug delivery should be considered and we are making effort to find some natural compounds which have the ability to activate Akt in vivo and to demonstrate the protective effects of these compounds in mice PD models.
Reviewer 2 Report
By using both an in-vitro and an in-vivo models, this study offers an underlying mechanism for rotenone neurotoxicity, specifically the Akt/mTOR pathway. In addition to post-exposure observations, the researchers establish this theory by manipulating this pathway and discussing the predicted results.
Some minor grammatical issues:
in section 2.5 – “autophary” should be autophagy discussion, page 7 – “activities of certain enzymes were significantly”Author Response
Please see the attachment.

Reviewer 3 Report
Zhang et al present a study examining how rotenone might confer neurodegeneration in Parkinson's disease. There are a number of grammatical and typographic errors in this manuscript which need addressing.
Major comments.
It is not clear to this reviewer why the protein phosphorylation study was not performed in mouse tissue. The mouse tissue data seems to just repeat previous experiments showing how rotenone is capable of causing behavioural changes and neurodegeneration in mice. The paper then switches to using SHSY5Y cells but it would seem that the two models are not linked in anyway especially given that the cell line is a human one. I would just have included the data from the SHSY-5Y cells with an expansion of some aspects.
The EM images included in figure 5 are not clear. I am unable to see the scale bar on this images, and because they are relatively low resolution, it is hard to determine whether the vacuolar changes are related to the rotenone treatment or are an artefact of the fixation and treatment of the cells prior to this.
Table 1 needs a better legend to explain what the values represent. For example the authors state that AKT ' was not detected after
rotenone treatment' however table 1 gives an average R value of 996283333. What does this value represent? how can this be described as not detected. Clarification would help readers who are unfamiliar with this technique.
Figure 3C is very unspecific in its labelling of the pathways for example 'organelle' and 'binding' are very ambiguous and could do with better descriptive terms.
Author Response
Reviewer 3
Zhang et al present a study examining how rotenone might confer neurodegeneration in Parkinson's disease. There are a number of grammatical and typographic errors in this manuscript which need addressing.
Major comments.
It is not clear to this reviewer why the protein phosphorylation study was not performed in mouse tissue. The mouse tissue data seems to just repeat previous experiments showing how rotenone is capable of causing behavioral changes and neurodegeneration in mice. The paper then switches to using SHSY5Y cells but it would seem that the two models are not linked in anyway especially given that the cell line is a human one. I would just have included the data from the SHSY-5Y cells with an expansion of some aspects.
Response: Thank you for the reviewing. Protein phosphorylation is usually an instant change in response to a stimulus. In this study, the establishment of the mouse PD model with rotenone injection was a long-term process and the mouse tissue seemed to be not appropriate for phosphorylation analysis. The human neuroblastoma cell line, SH-SY5Y, is a commonly used cell line in studies related to neurodegenerative diseases. Systems genomics approach showed that most of the genes belonging to the major Parkinson’s disease pathways and modules were intact in the SH-SY5Y genome. Specifically, each analyzed gene related to PD had at least one intact copy in SH-SY5Y cells (Systems genomics evaluation of the SH-SY5Y neuroblastoma cell line as a model for Parkinson’s disease. Krishna et al. BMC Genomics 2014, 15:1154). Thus, we chose SH-SY5Y cells instead to establish a PD model in vitro and to do the proteomic analysis.
The EM images included in figure 5 are not clear. I am unable to see the scale bar on this images, and because they are relatively low resolution, it is hard to determine whether the vacuolar changes are related to the rotenone treatment or are an artefact of the fixation and treatment of the cells prior to this.
Response: The seemingly low resolution of the images may be due to the compression. According to your suggestion, we included a new scale bar and higher magnifications of pathological mitochondria in the new Figure 5.
Table 1 needs a better legend to explain what the values represent. For example the authors state that AKT ' was not detected after rotenone treatment' however table 1 gives an average R value of 996283333. What does this value represent? how can this be described as not detected. Clarification would help readers who are unfamiliar with this technique.
Response: Sorry for the confusion caused by not describing the details in Table 1. We added the explanation in the revised version. The average C value of 996283333 for AKT2 was the intensity of phosphorylated AKT in Control (C) group while it was not detectable (-) in Rotenone (R) group.
Figure 3C is very unspecific in its labelling of the pathways for example 'organelle' and 'binding' are very ambiguous and could do with better descriptive terms.
Response: Thank you for the suggestions and we replaced figure 3C with a new one.
Round 2
Reviewer 1 Report
The authors' response to the comments are generally well thought except for point (2). There are well established information regarding AKT activation and neuroprotection
(Neuroprotective Role of the PI3 Kinase/Akt Signaling Pathway in Zebrafish - PMID: 28228749)
(Resveratrol Suppresses Rotenone-induced Neurotoxicity Through Activation of SIRT1/Akt1 Signaling Pathway - PMID: 29350822).
(SC79, a novel Akt activator, protects dopaminergic neuronal cells from MPP+ and rotenone. PMID: 31342299)
Authors should include more information that reinforce the neuroprotection idea and discuss the limitation of their study, once they did not tested in vivo.
Author Response
Thank you for the advice and we have included the discussion in the revised version.
Reviewer 3 Report
I thank the authors for their responses to my points.
With regards to my first point regarding the inclusion of data from both mice and SHSY5Y cells, I think the authors may have missed my point. It is clear that both rotenone treated animals and SHSY5Y cells are both routinely used models for PD, though both have their flaws. I was not asking for justification of the use of these models, as both are appropriate. Rather it did not seem to make sense to me to include data from the mouse model, which does not show anything novel, if these tissues from these animals were not then used in further phosphorylation studies. If the authors did want to include this data there needs to be a better description of why these tissues were not then used in further studies, as briefly mentioned in their rebuttal (‘the tissue seemed not to be appropriate for phosphorylation analysis’). I however would still be tempted to remove the mouse data and focus solely on the data collected from the SHSY5Y cells.
The changes made to figure 3 do improve its clarity.
The changes made to figure 5, also improve it.
Table 1 has also been improved.
Author Response
Thank you for the positive assessment of the revised version. According to your suggestion, we have included the explanation of the mouse data.